# The effects of department of Veterans Affairs medical centers on socio-economic outcomes: Evidence from the Paycheck Protection Program

**Christos A. Makridis**[1,2,3]*, **J. D. Kelly**[3], **Gil Alterovitz**[1,4]

**1** National Artificial Intelligence Institute, Department of Veterans Affairs, Washington, District of Columbia, United States of America, **2** Columbia Business School, New York, NY, United States of America, **3** Stanford University, Stanford, California, United States of America, **4** Harvard Medical School, Cambridge, Massachusetts, United States of America

* christos.makridis@va.gov

**Data Availability Statement:** All files are available from the following link: https://github.com/cmakridis777/PLOS-One-Veteran-PPP-Data.

## Abstract

Do medical facilities also help advance improvements in socio-economic outcomes? We focus on Veterans, a vulnerable group over the COVID-19 pandemic who have access to a comprehensive healthcare network, and the receipt of funds from the Paycheck Protection Program (PPP) between April and June as a source of variation. First, we find that Veterans received 3.5% more loans and 6.8% larger loans than their counterparts ($p < 0.01$), controlling for a wide array of zipcode characteristics. Second, we develop models to predict the number of PPP loans awarded to Veterans, finding that the inclusion of local VA medical center characteristics adds almost as much explanatory power as the industry and occupational composition in an area and even more than the education, race, and age distribution combined. Our results suggest that VA medical centers can play an important role in helping Veterans thrive even beyond addressing their direct medical needs.

## Introduction

While the COVID-19 pandemic has affected everyone, some groups were at greater risk than others. United States' Veterans, for example, tend to be older than the average American (58 versus 38 years old) And yet, they displayed slightly lower rates of infection than their counterparts and better economic outcomes over the pandemic [1]. In particular, as of Oct. 22, the incidence in the military was 2,387 cases per 100,000 (52,321 cases in a population of 2,191,000, active and reserve), whereas it was 2,527 cases per 100,000 (8,338,000 cases in a population of 330 million). Given that Veterans also have access to the largest, integrated healthcare services network in the country through the Department of Veterans Affairs, a natural question is whether access to medical services helped cushion against the economic effects of the pandemic.

Small businesses have been among the most disproportionately affected by the COVID-19 pandemic [2]. Motivated by an urgency to help businesses cope with the national and state

**Funding:** The authors received no specific funding for this work.

**Competing interests:** Gil and Christos completed this research through their roles on the National Artificial Intelligence Institute at the Department of Veterans Affairs. J. D. completed this research during his undergraduate studies at Stanford University.

quarantines, Congress united to pass the Paycheck Protection Program (PPP) in April 2020. However, given the decline in entrepreneurship rates among Veterans [3], coupled with evidence Veterans increasingly reside within cities with lower wage, employment, and per capita productivity growth [4], it is possible that Veterans might be more exposed to the adverse effects of the pandemic and less likely to receive support from the PPP. Using recently-available data from the Small Business Administration (SBA) on all the PPP loans under $150,000, this paper investigates how Veterans fared, relative to their counterparts, and the role of VA medical centers in explaining these differences.

In the first part of the paper, we show that Veterans received 3.5% more loans and 6.8% larger loans than their counterparts. Our statistical strategy controls for a wide array of zipcode characteristics, such as age and education, and exploits within-zipcode variation in further robustness. That allows us to compare differences in PPP receipt among Veterans and non-Veterans within the same zipcode over time. These results are important given that minority and rural borrowers did not receive as many loans as their counterparts [5]. That minorities and rural borrowers received fewer PPP loans in the raw data, however, could reflect selection effects. In particular, [6] find that banking relationships that existed prior to the pandemic among businesses are predictive of early access to PPP funds. Since minority and rural borrowers are less likely to have pre-existing banking relationships, naturally they will have lower PPP receipt.

In the second part of the paper, we use an ensemble of machine learning techniques to understand the features that predict differences in the receipt of PPP among Veterans. In addition to standard demographic factors, including the share of Veterans within a zipcode, we gather new data on the distance to the nearest Department of Veterans Affairs (VA) medical center and the medical center's quality, which we obtain through new data on patient ratings. While demographic factors tend to matter most in predicting PPP loan volume for Veterans, characteristics about the local VA medical centers also matter, especially proxies for quality (e.g., wait time, VA patient satisfaction scores). We also show that VA medical center characteristics are not just absorbing variation with potential correlates, but rather adding meaningful predictive power to these models.

Our paper is closely related with an emerging empirical literature on the effects of the PPP on economic outcomes, as well as a larger literature on the provision of liquidity during crises. For example, [7] conduct a survey of small businesses during the roll-out of the PPP and find that many were liquidity constrained and uncertain about whether they would withstand the pandemic. These considerations prompted roughly 70% of respondents to anticipate taking advantage of the PPP to avoid laying off employees and sustaining operations. Similarly, [8] conduct a survey on small businesses, focusing on the information frictions that can emerge and stifle the allocation of credit. [9] find that the PPP boosted employment at eligible firms by 2–4.5%, whereas [10, 11] find that the short to medium -term effects were more muted. These recent explorations of the PPP are connected with a larger literature on government intervention in credit markets. This paper also provides clarity on the heterogeneous effects of public policy. While there is anecdotal evidence of heterogeneity in the receipt of PPP among minorities, there is not yet any evidence for Veterans, despite their substantial role in the U.S. population.

Our paper is also related with the socio-economic determinants of health [12, 13]. For example, [14, 15] investigate the socio-demographic correlates of COVID-19 mortality among veterans, documenting differences across race, education, and income. Others have emphasized the role of other co-morbidities, like asthma, as risk factors for Covid-19 [16]. Similarly, [17] show how social capital—the degree of shared norms, trust, and networks—within a county helped shield against infections and mortality from the virus. Our paper complements

these by providing the first quantitative evidence on the role of local VA medical centers and how their quality can cushion against co-morbidities and other local factors that may affect transmission.

Relatedly, there has been some study of the effects of community programs on public health. For example, [18] discusses a community health center in the Mississippi Delta that "created programs designed to move beyond narrowly focused disease-specific interventions and address some of the root causes of community morbidity and mortality," including social and educational opportunities. More recently, [19] explore the role of community health centers (CHCs) over the COVID-19 pandemic, focusing on their specific experience at the Mass General Brigham Hospital. In this sense, our result on the predictive power of higher quality VAMCs is consistent with this prior literature on CHCs.

## Data and measurement

There is now an emerging literature on the Paycheck Protection Program (PPP). The following summarizes its institutional details here [6]. The PPP began on April 3rd, 2020 from the 2020 CARES Act as a temporary source of liquidity for small businesses, authorizing $349 billion in forgivable loans to help small businesses pay their employees and additional fixed expenses during the COVID-19 pandemic. The Small Business Administration (SBA) was responsible for overseeing firm applications through banks, focusing on increasing access to credit among small businesses with 500 or fewer employees with some exceptions.

While small businesses were eligible as of April 3rd, independent contractors and self-employed workers became eligible shortly after as of April 10th. The terms of the loan are the same for all businesses set such that the maximum amount of a PPP loan is the lesser of 2.5 times the average monthly payroll costs or $10 million where the average monthly payroll is based on prior year's payroll after subtracting the portion of compensation to individual employees that exceeds $100,000. While there was uncertainty abut the terms of the program, businesses would have their loans forgiven if (i) the loan proceeds were used to cover payroll costs, mortgage interest, rent, and utility costs over the eight-week period following the provision of the loan (and not more than 25 percent of the loan forgiveness amount may be attributable to non-payroll costs), and (ii) employee counts and compensation levels were held fixed.

We use micro-data that was made available through the Small Business Administration and the Department of Treasury containing all the PPP loans. We focus on the loans that were made under $150,000 since these are more likely to be directed towards small businesses, specifically Veterans. The SBA PPP Loan dataset presents data on the loan amount, location, type of business, business owner demographics, job retention, lender, and date of approval for all active and approved PPP loans awarded by the SBA. However, since demographic data was provided to lenders, and subsequently the SBA, on a voluntary basis, a large sample of our observations are missing information about not only race (80% missing), but also Veteran status (75%). Nonetheless, the data is still highly informative for our predictive modeling, especially after we control for local characteristics. We observe 26,174 observations for Veteran-owned businesses and 574,977 of their counterparts. From this data, we aggregate all loans at the zipcode level to present the number of loans, the total amount of loans, jobs retained, and the number of loans awarded to each business type and owner demographic for all zipcodes represented. The contribution to each of these measures by reported Veteran-owned businesses is also presented in the extracted features.

The S1 Appendix presents two sets of results to assess the representativeness of our data on Veterans. First, we regress household income, housing values, and the unemployment rate on an indicator for whether the zipcode ranks above the median with respect to missing values of

Veteran status in the PPP data. When we weight by the number of respondents in a zipcode, we find that there are only economically insignificant, and sometimes statistically insignificant too, differences between these sets of zipcodes, suggesting that measurement error in Veteran status is not too large. Second, we show that the share of Veterans implied by the PPP data has a 0.30 correlation with the share of Veterans obtained directly from the Census within a zipcode, suggesting that we are capturing the general direction of the data.

Lastly, we include data on the quality of care at and relative location of the closest VA Medical Centers. Using the VA facilities API we are able to query the most recent data and metrics on the closest VA Medical Centers in order to evaluate their services. We provide data on two primary types of VA Medical Centers for each zipcode. Often smaller VA clinics provide local communities with primary care in addition to larger regional VA Medical Centers with a broader array of primary and specialty medical services. We first include data on the average drive time to the closest VA medical center, distance to the medical center, patient satisfaction scores for routine primary care, and average wait time for established primary care. Secondly, we include the distance, patient satisfaction scores for primary and specialty care in both routine and urgent cases, and average wait time for new and established primary and specialty care at the closest large regional VA Medical Center. The open-source GeoPy geocoding library is used to approximate the distance between zipcodes and nearby VA facilities. It is possible that both locations are the same if the closest center to a given zipcode is a large regional VA Medical Center. Since an individual in any given community could interact with both smaller local and larger regional medical centers for care, both centers impact the economic ecosystem in each zipcode. See The S1 Appendix for a description of all the variables.

In order to perform regression on the dataset, the data required pre-processing and cleaning. We first eliminated uncommon features within the dataset. This was done to reduce the number of samples with missing features. Analysis was performed to ensure eliminated features were randomly distributed within samples, and that the data within the eliminated features had minimal contributions to regression predictions. Once uncommon features were eliminated from the dataset, samples with missing features from those remaining were also eliminated.

The final clean dataset contained 157 features and 22962 zipcode samples. We should note that over the course of our analysis not all features were used in every instance, however, the samples remained consistent. From the remaining samples, feature interaction was performed between zipcode demographics features and VA Facility features to account for non-linear interactions between theses features within a sample. Potential target vector columns were removed from the data set and stored for future prediction training, validation, and testing. The primary target values were the number of loans awarded to Veteran owned businesses in each zipcode. The features from the dataset were then shifted and scaled such that each feature column was centered around zero and had a variance in the same order. Additionally, an offset one-vector was included to accommodate the simplest linear regression models, however, this feature column will effectively be ignored by more complex models.

## Did the Paycheck Protection Program help Veterans?

We begin by investigating whether Veterans received higher proportions of PPP loans and their relative loan volume. That is, conditional on receiving a PPP loan, we compare how Veterans fare relative to their non-Veteran counterparts through regressions of the form:

$$y_{izt} = \gamma VET_{izt} + g(X_z, \theta) + \phi_l + + \xi_g + \lambda_t + \epsilon_{izt} \tag{1}$$

**Table 1. Baseline results on loan amount and Veteran status.**

| Dep. var. = | log(Number of Loans) | | log(Loan Amount) | | | | |
|---|---|---|---|---|---|---|---|
| Is Veteran × Information Sector | -.055 *** [.016] | .035 *** [.011] | .105 *** [.009] | .108 *** [.008] | .095 *** [.008] | .068 *** [.008] | .068 *** [.008] .012 [.035] |
| R-squared | .00 | .12 | .00 | .00 | .17 | .22 | .22 |
| Sample Size | 601151 | 600720 | 601147 | 600716 | 597282 | 587090 | 587090 |
| Zip Controls | No | Yes | No | Yes | No | No | No |
| Zip FE | No | No | No | No | Yes | Yes | Yes |
| Industry FE | No | No | No | No | No | Yes | Yes |
| Time FE | No | No | No | No | Yes | Yes | Yes |

Notes.—Sources: Small Business Administration Paycheck Protection Program (PPP) and 2014–2018 American Community Survey. The table reports the coefficients associated with regressions of the logged number of loans in a zipcode × day logged loan amount on an indicator for being a Veteran, conditional on zipcode controls. Controls include: logged population, the age distribution (the share of individuals under age 18, 18–24, 45–64, 65+), the share married, the education distribution (the share with less than a high school degree, some college, and college plus), the share of Veterans ages 18–64, and the share of Veterans over the age of 65. Standard errors are clustered at the zipcode-level and observations with the weight are using the number of respondents in the zipcode.

where $y$ denotes the outcome variable of interest for individual $i$ in zipcode $z$ and time period $t$, $VET$ denotes an indicator for whether an individual is a Veteran, $g(X, \theta)$ denotes a semi-parametric function of zipcode-level controls, $\phi$ denotes fixed effects on location (e.g., county or zipcode), $\xi$ denotes fixed effects on the group (e.g., three-digit industry) and $\lambda$ denotes fixed effects on day-of-the-year. Standard errors are clustered at the zipcode-level.

Table 1 documents the results under several specifications. Starting with the logged number of PPP recipients within a zipcode × day, column 1 shows the raw correlation with Veteran status, suggesting that they receive 5.5% more PPP loans in a location. However, one concern with the raw correlation is that differences in banking networks could be correlated with differences in where Veterans decide to live. After we control for our zipcode-level controls, including the share of Veterans, we find that Veterans receive 3.5% more loans within a location.

Turning towards the logged loan amount, conditional on receiving a loan, we find that Veterans receive larger loans, on average. While the raw correlation suggests that Veterans earn 10.5% larger loans (column 1), the correlation drops to 9.5% when we add zipcode and time fixed effects (column 3) and 6.8% when we add three-digit NAICS fixed effects (column 4). This captures the fact that certain industries were more adversely affected by the pandemic due to, for example, differences in their digital intensity [20], which could be correlated with their demand for liquidity. We also allow for an interaction between Veteran status and an indicator for whether or not the business is in the information services sector. While our estimate is statistically insignificant, we find that Veterans in these sectors received 1.2% larger loans.

## Understanding differences in Veteran receipt of PPP

We evaluate four ML techniques in order to derive the significance of zipcode level features on the number of PPP loans awarded in each zip code. We use ordinary least-squares regressor, ridge regressor, support vector regressor (SVR) with a linear kernel, and XGBoost Regressor models. A critical aspect of our analysis of feature significance is the ability to easily ascertain

the influence of each feature on the trained model. Previous studies have shown this to be a useful analytical tool to understand the data used to train these models [21].

The need for this capability limited the number of models well-suited for the study. While it is possible more complex or other nonlinear models could outperform those selected, and there exist some tools to address the interpretability of such models, we stress that the goal of this study is to leverage ML models to better understand the data and not to develop the most accurate regressors possible [22]. Therefore, we do not anticipate that the exclusion of additional models in favor of those that are more easily interpretable will significantly impact results.

The first models we evaluate are ordinary least-squares and ridge regression models. These linear models benefit from simplicity and short training time. However, they are constrained to predict based solely on linear relationships between the input features and prediction [23, 24]. Ridge regression implements regularization, allowing for greater flexibility than standard least squares estimators [24].

We also evaluate an SVR with linear kernel model. SVR models are more complex than those previously discussed, which can increase dramatically the time it takes to train the classifier since the training set scales [25]. The SVR with a linear kernel is far better suited to quickly train on larger datasets as opposed to non-leaner kernel variants [26]. Furthermore, using a linear kernel allows us to better interpret feature importance directly from feature coefficients, which was a primary constraint in our model selections [26].

Lastly, we evaluate an Extreme Gradient Boost ("XGBoost Regressor"), which leverages weaker learning algorithms in a tree structure to derive a stronger learning algorithm that is well suited for complex regression tasks and large datasets [27]. A particularly useful component of the XGBoost Regressor is the feature importance attribute that states the significance of each feature's contribution to the prediction [27]. (The ordinary least-squares, ridge, and SVR models were implemented using the open-source scikit-learn python module [28]. The XGBoost Regressor model was implemented with a separate open-source python module that supports a scikit-learn wrapper [27]).

We developed a regression model for each technique to perform regression on the number of loans granted to Veteran-owned businesses in each given zip code. Hyperparameters were optimized through an exhaustive grid search with five-fold cross-validation. In order to ensure consistency across datasets, the same hyperparameters are used for all models trained with a given technique. This is determined by the mode of optimized hyperparameter values. Table 2 presents the hyperparameters chosen for evaluation; they have expanded from those that have been shown to optimize similar problems [21]. We use the coefficient of determination ($R^2$) to

**Table 2. Hyperparameter configurations.**

| ML Models | Hyperparameters | Values |
|---|---|---|
| OLS Regressor | Fit Intercept | {True, False} |
| Ridge Regressor | Alpha | {0.0001, 0.001, 0.01, 0.1, 1} |
| Linear SVR Regressor | C | {1, 10, 100} |
| XGBoost Regressor | Learning Rate | {0.01, 0.1, 0.2} |
| | Min Child Weight | {1, 3, 5} |
| | Max Depth | {3, 6, 9} |
| | Number of Estimators | {500} |

Notes.— The table reports the hyperparameters evaluated to identify the best performing configurations through an exhaustive grid search ob each model

**Table 3. ML model $R^2$ performance metrics over all datasets.**

| Dataset = | Zipcode Demographics | VA Facilities | Zipcode and VA | VA-Zipcode Interactions |
|---|---|---|---|---|
| OLS Regressor | 0.394 (0.013) | 0.120 (0.004) | 0.397 (0.015) | 0.437 (0.021) |
| Ridge Regressor | 0.394 (0.012) | 0.120 (0.004) | 0.397 (0.015) | 0.440 (0.018) |
| Linear SVR Regressor | 0.323 (0.018) | -0.218 (0.007) | 0.324 (0.011) | 0.380 (0.016) |
| XGBoost Regressor | 0.477 (0.019) | 0.241 (0.015) | 0.484 (0.027) | 0.494 (0.021) |

Notes.—Sources: Small Business Administration Paycheck Protection Program (PPP), 2014–2018 American Community Survey, and Department of Veterans Affairs Facilities API. The table reports the coefficient of determination ($R^2$) performance metric for trained models over each dataset and ML technique. Standard deviations are presented in the parenthesis. The Zipcode Demographics Dataset features contain only zipcode level demographic data. The VA Facility Dataset contains features only pertaining to the access and quality of care of the closest local and large regional VA Medical Centers. The Zipcode and VA Dataset contains all of the features from the first two datasets. The Zipcode-VA Interaction Dataset includes all features from the third dataset and new feature columns that interact each feature related to VA facility access and quality of care with each demographic level zipcode feature.

evaluate the performance and error of each regressor model. Once the optimal hyperparameters were determined, each model was trained ten times using randomly partitioned 80–20 training-testing splits. Testing samples are used to determine performance evaluations, and the large number of trials contributes to confidence in model performance evaluations.

To understand the influence of our features on model performance and predictive capabilities, we develop models for multiple different sets of features all predicting the number of PPP loans granted to Veteran owned-businesses in a zip code. The progressive datasets allow us to understand the predictive power of each group of features and how the interaction between different features, indicative of a VA facility's impact on its surrounding community, contributes to model performance. We restrict each dataset to the same zipcodes and perform regressions on the same target vector, with the different features mentioned, to ensure comparability is maintained.

We train models with optimal hyperparameters using each of our four machine learning techniques over all of the datasets 10 times. During each run, the coefficient of determination and root mean squared error are recorded in order to comparatively evaluate model performances. In Table 3 we present the average $R^2$ with standard deviation (in parenthesis), respectively, for all models trained on each dataset. We see that for all datasets, the ranking of model performance is consistent. All four of these regression problems are fairly similar, and as such it is not surprising that we see similar trends in performance.

While the model with demographic and VA medical center interactions performs the best with an $R^2$ between 0.437 and 0.494 depending on the model, the biggest gains in predictive power comes from our use of XGBoost, rather than, say, linear regression. For example, whereas linear regression produces an $R^2$ of 0.394 with demographics only, XGBoost produces an $R^2$ of 0.477. Similarly, the $R^2$ climbs from 0.437 to 0.494 in our model with demographic and VA medical center interactions. We also see that VA medical center characteristics alone are not highly predictive, generating $R^2$s under 0.24. One reason for these results is that medical centers will play little role in advancing entrepreneurship if the population is not likely to select into entrepreneurship in the first place (e.g., older residents), so incorporating both VA medical center and demographic characteristics maximizes their predictive power.

Another approach to gauge the performance of these models is the Regression Receiver Operating Characteristic (RROC) curve plot. Similar to the Receiver Operating Characteristic (ROC) curve plot for classification models, RROC plots allow us to visualize the relative accuracy of regression models [29]. Additionally, RROC plots include over and underestimation by

regressors rather than by an absolute tolerance [29]. This information presents bias in the model predictions. The bias is indicated by the box marked with an "x" on each curve. If this falls to the left of the dashed-diagonal line on the plot the regressor is biased towards overestimation, while falling on the left indicates bias towards underestimation. An ideal model would resemble a right angle in the upper left-hand corner of the plot with the box indicating bias falling on the dashed-diagonal, as all predictions would have neither overestimation nor underestimation and there would be no bias. The best real models will approach this visualization. In Fig 1 we present the corresponding RROC curves for the XGBoost Models trained on each dataset.

In addition to comparing the performance metrics of models with and without various features derived from VA Medical Center Data and Zipcode Demographics, we also evaluate the significance of features to the predictive outcomes in models that incorporate Zipcode level Demographics and VA Facility Data. This is applicable to the Zipcode and VA Facility Interaction dataset and the Zipcode-VA Facility Interaction dataset as both include these features. XGBoost Regressor models have a callable attribute "feature importances" that ranks each

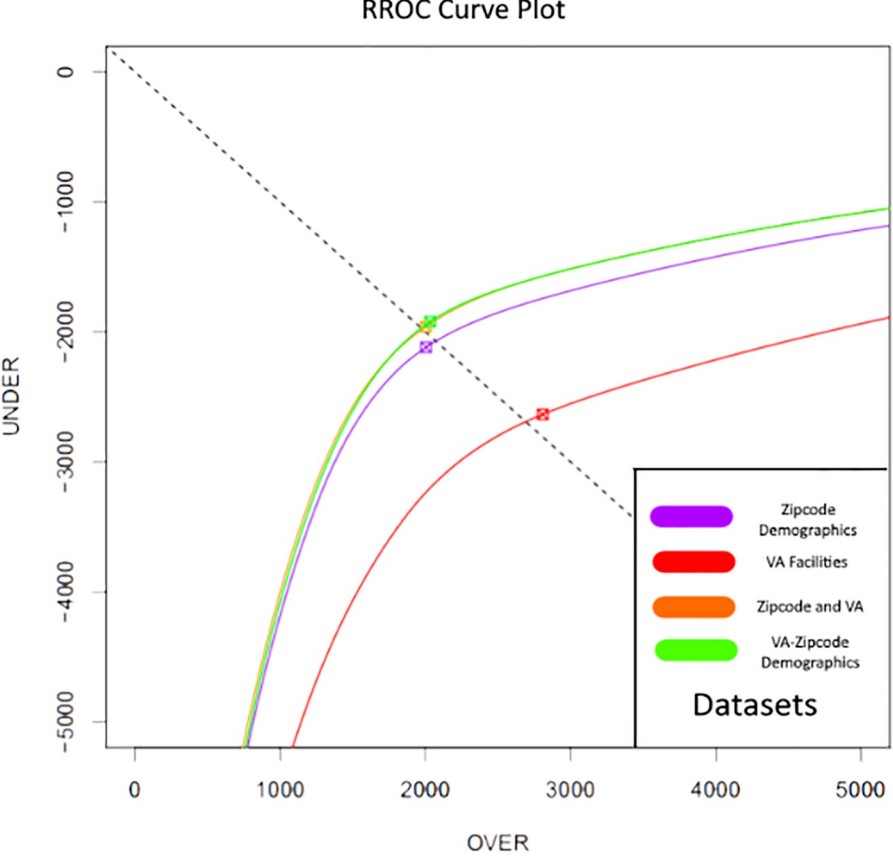

**Fig 1. RROC curve for XGBoost model trained on each dataset.** Notes.—The figure plots the RROC curve for XGBoost Models trained on 4 datasets. The Zipcode Demographics Dataset features contain only zipcode level demographic data. The VA Facility Dataset contains features only pertaining to the access and quality of care of the closest local and large regional VA Medical Centers. The Zipcode and VA Dataset contains all of the features from the first two datasets. The VA-Zipcode Interaction Dataset includes all features from the third dataset and new feature columns that interact each feature related to VA facility access and quality of care with each demographic level zipcode feature. RROC plots visualize the relative accuracy of regression models as well as indicating bias in the model by plotting with respect to over and under estimation of predictions [29].

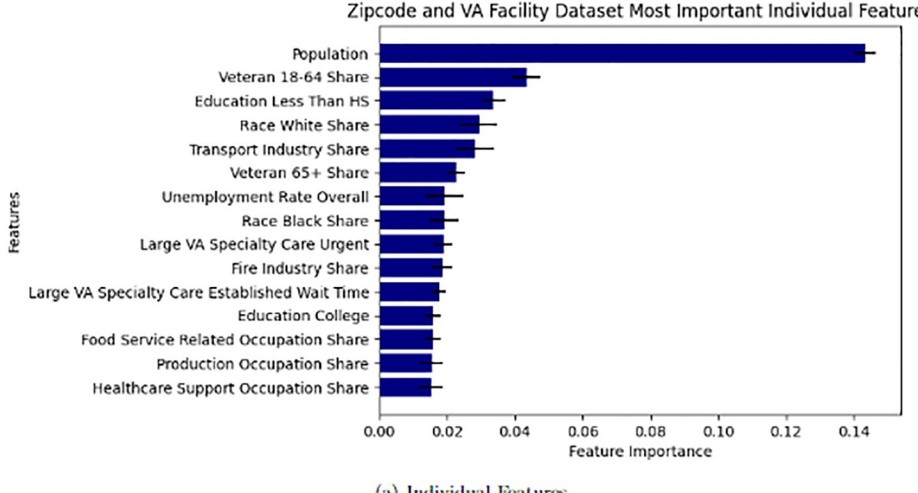

(a) Individual Features

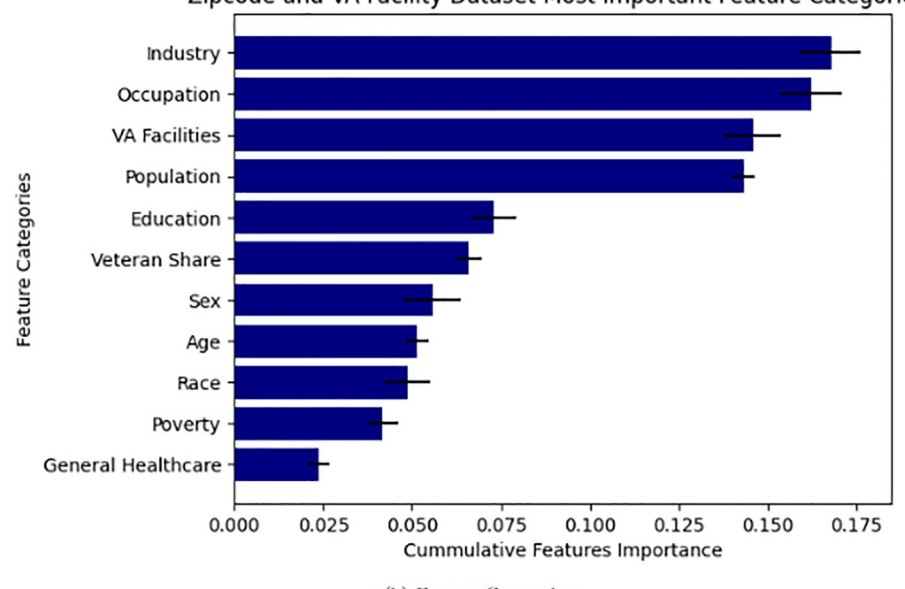

(b) Feature Categories

**Fig 2. Feature importance for predicting PPP loans to Veterans.** Notes.— Feature Importances are given by an XGBoost model trained on the Zipcode Demographics and VA Facility Dataset. The dataset includes features pertaining to zipcode level demographic data, and access and quality of care of the closest local and large regional VA Medical Centers. The Individual Feature figure plots the 15 most significant individual features by value. The Feature Category figure plots the importance of Feature Categories by the cumulative importance of the features in each category. The error bars represent one standard deviation from the mean importance. A: Individual Features. B: Feature Categories.

feature's influence on the prediction of the model in the form of a decimal value from 0 to 1. In the Figs 2 and 3 we present the 15 most significant individual features with their respective feature significance value as well as feature categories with each category's cumulative feature significance for both datasets. The error bars represent one standard deviation from the mean for each average value.

Not surprisingly, population is the most significant predictor, which reflects the fact that areas with more people have a greater demand for credit and, therefore, more loan applications

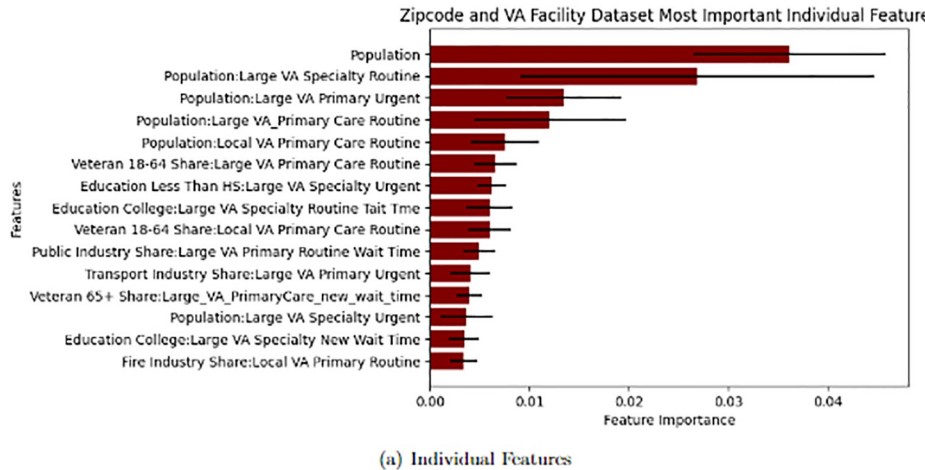

(a) Individual Features

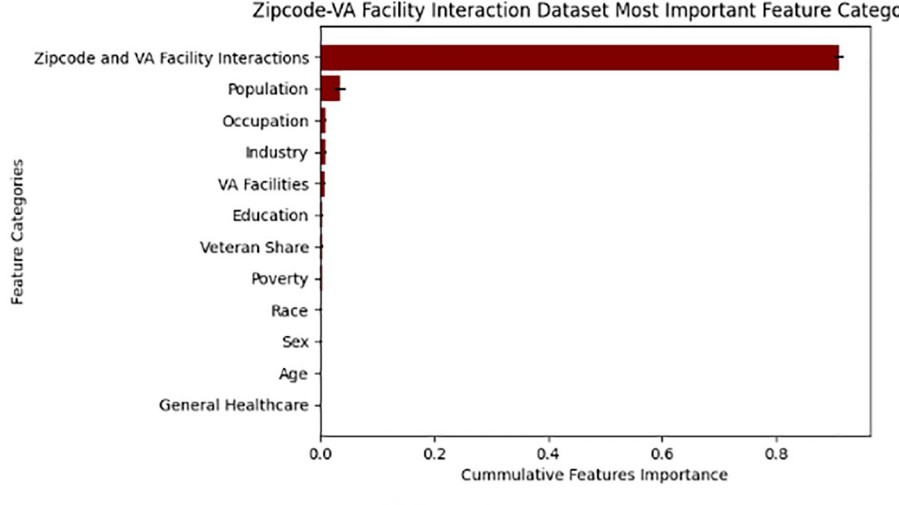

(b) Feature Categories

**Fig 3. Zipcode-VA facility interaction importance for PPP loans to Veterans.** Notes.— Feature Importances are given by an XGBoost model trained on the Zipcode Demographics—VA Facility Interaction Dataset. The dataset includes features pertaining to zipcode level demographic data, access and quality of care of the closest local and large regional VA Medical Centers, and feature columns that interact each feature related to VA facility access and quality of care with each demographic level zipcode feature. The Individual Feature figure plots the 15 most significant individual features by value. The Feature Category figure plots the importance of Feature Categories by the cumulative importance of the features in each category. The error bars represent one standard deviation from the mean importance. A: Individual Features. B: Feature Categories.

and eventual receipts. We also find that zipcodes with a higher share of Veterans are more likely to receive Veteran PPP loans, which operates as an internal validation mechanism. We also find that the education, race, and industrial composition are highly predictive. Interestingly, however, VA medical center characteristics, such as the quality of specialty care and its corresponding wait time, emerge are predictive. Fig 3 summarizes the results by taking the sum of individual features across different categories of variables. While industry and occupational composition matter the most, VA facility characteristics emerge as a close runner-up, explaining roughly 15% of the variation in PPP loan outcomes. Importantly, VA characteristics matter more than other demographic characteristics, such as education, race, age, gender, and even poverty rates.

**Table 4. Model $R^2$ performance metrics over all subgroups and datasets.**

| Dataset = | Zipcode Demographics | VA Facilities | Zipcode and VA | VA-Zipcode Interactions |
|---|---|---|---|---|
| Large Veteran Share 65+ | 0.490 (0.028) | 0.227 (0.024) | 0.481 (0.038) | 0.511 (0.025) |
| Small Veteran Share 65+ | 0.438 (0.017) | 0.215 (0.024) | 0.441 (0.025) | 0.445 (0.027) |
| Large Veteran Share 18–64 | 0.497 (0.025) | 0.227 (0.015) | 0.51 (0.022) | 0.514 (0.017) |
| Small Veteran Share 18–64 | 0.417 (0.034) | 0.225 (0.031) | 0.443 (0.022) | 0.423 (0.05) |
| Close to Large VA Facility | 0.448 (0.018) | 0.199 (0.021) | 0.445 (0.029) | 0.455 (0.023) |
| Far from Large VA Facility | 0.451 (0.022) | 0.203 (0.026) | 0.445 (0.030) | 0.456 (0.021) |
| Close to Local VA Facility | 0.432 (0.024) | 0.190 (0.024) | 0.427 (0.038) | 0.427 (0.038) |
| Far from Local VA Facility | 0.399 (0.028) | 0.037 (0.026) | 0.397 (0.026) | 0.416 (0.036) |

Notes.—Sources: Small Business Administration Paycheck Protection Program (PPP), 2014–2018 American Community Survey, and Department of Veterans Affairs Facilities API. The table reports the coefficient of determination ($R^2$) performance metric for trained models over each dataset and ML technique. Standard deviations are presented in the parenthesis. The Zipcode Demographics Dataset features contain only zipcode level demographic data. The VA Facility Dataset contains features only pertaining to the access and quality of care of the closest local and large regional VA Medical Centers. The Zipcode and VA Dataset contains all of the features from the first two datasets. The Zipcode-VA Interaction Dataset includes all features from the third dataset and new feature columns that interact each feature related to VA facility access and quality of care with each demographic level zipcode feature. These subgroups explore samples with values above and below the median for each of the following features: Share of Veteran Population 18—64, Share of the Veteran Population 65 and older, Distance to a Local VA Facility, and Distance to a Large Regional VA Facility.

We now turn towards more flexible interactions between demographic and VA characteristics in Fig 3. We find that the addition of VA Facility features and the interaction between VA Facility features and Zipcode Demographic features contribute to greater $R^2$ values. This is indicative of better performing models, however, we note that the performance gain is small. Additionally, we see that trained models place respectively high importance on features related to VA facilities in the third and fourth dataset when compared to other feature categories.

To better understand the impact of VA facilities on various communities where Veteran's reside, and how these facilities may have impacted propensity to receive PPP loans, we further train models on eight subgroups of samples exploring the share of Veteran population 18—64, share of the Veteran population 65 and older, distance to a local VA Facility, and distance to a large regional VA Facility. Using the previously optimized hyperparameters, we train four XGBoost models with features from each of the established datasets on samples from each subgroup. These models are trained 10 times, presenting the average and standard deviation $R^2$ on each model in Table 4. Building on the results from Table 3, we see that our predictive models perform better in areas with higher shares of Veterans, regardless of whether we measure the Veteran share based on the share between age 18–64 or 65+. For example, the $R^2$ in the model with interactions is 0.511 on the sample of zipcodes above the median share of Veterans and 0.445 on the sample below the median share. We find no difference in the $R^2$ among zipcodes that are closer versus further away from VA medical centers.

## Discussion and interpretation of results

Our results highlight the importance of VA medical centers as predictors for Veteran outcomes under the PPP. However, one of our concerns is that VA medical center characteristics simply behave as a proxy for other omitted variables that are correlated with both PPP loan receipt and demographic characteristics. If that were the case, then our gains in model performance might spuriously reflect VA medical characteristics when really there are other fundamental sources of the PPP loan disbursement.

While concerns about omitted variables are always present and impossible to fully rule out, we nonetheless address the possibility by estimating new predictive models that embed variables that potentially proxy for omitted characteristics. We focus on social capital characteristics at the county-level, produced by the Joint Economic Committee in their social capital project [30], which includes measures of family unity, community health, institutional health, and collective efficacy.

We previously presented feature importance data to demonstrate the role VA Facility related features played in our trained models' predictions. Once again we employ feature importance derived from an optimized XGBoost model to demonstrate that VA Facility related features contribute unique data to our model. The model was trained 10 times and averaged, the error bars represent one standard deviation from the mean for each averaged value. Fig 4 presents feature categories as previously defined with the addition of new features under the "Omitted Variables" classification. While our omitted variables have a cumulative feature importance of 0.036 (out of 1 with a std of 0.003), our VA Facility related features have a cumulative feature importance of 0.144 (std of 0.004). The fact that the previously omitted variables provide some additional predictive power is not surprising, but the important result is that the VA Facility related features continue to contribute significantly to model predictions. That means they must provide additional unique information beyond that of the previously omitted variables. We would expect a significant reduction in VA Facility feature importance from prior models (see Fig 2) if confounding omitted variables were represented through VA Facility data rather than specific and unique data about the VA Facilities themselves.

Given that these results illustrate that VA medical center characteristics matter, and are not simply proxies for other unobserved characteristics, we aim to better understand the potential

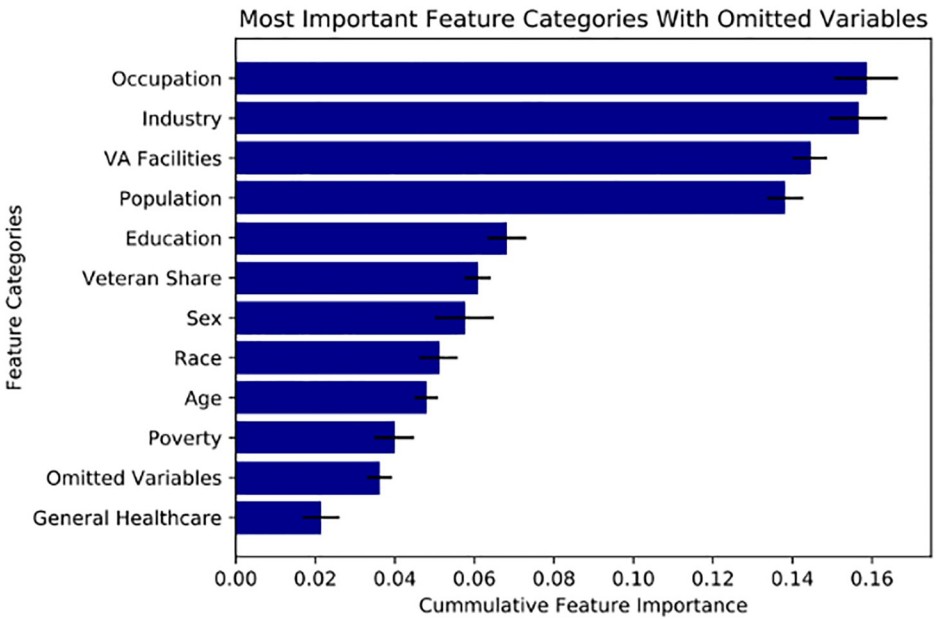

**Fig 4. Feature categories importance with omitted variables for PPP loans to Veterans.** Notes.— Feature Importances are given by an XGBoost model trained on the Zipcode Demarphics and VA Facility Dataset with new Omitted Variables. The dataset includes features pertaining to zipcode level demographic data, access and quality of care of the closest local and large regional VA Medical Centers, and new omitted varaibles. The Feature Category figure plots the importance of Feature Categories by the cumulative importance of the features in each category. The error bars represent one standard deviation from the mean importance.

**Table 5. Decomposing the determinants of VA center quality.**

| Dep. var. = | log(Overall VA Center Quality) | | | | | | | | |
|---|---|---|---|---|---|---|---|---|---|
| log(Communication w/ Doctors/Nurses) | 1.559 *** [.106] | | | | | | | | .173 [.160] |
| log(Care Transition) | | .932 *** [.069] | | | | | | | .085 [.094] |
| log(Hospital Cleanliness) | | | .662 *** [.064] | | | | | | .128 ** [.052] |
| log(Communcation about Medication) | | | | .735 *** [.102] | | | | | -.200 *** [.071] |
| log(Pain Management) | | | | | .683 *** [.115] | | | | .099 ** [.049] |
| log(Hospital is Quiet) | | | | | | .300 *** [.057] | | | .078 *** [.029] |
| log(Responsiveness of Staff) | | | | | | | .663 *** [.078] | | .062 [.050] |
| log(Willing to Recommend) | | | | | | | | .887 *** [.041] | .677 *** [.061] |
| R-squared | .71 | .71 | .67 | .54 | .56 | .49 | .61 | .89 | .93 |
| Sample Size | 122 | 122 | 122 | 122 | 122 | 122 | 122 | 122 | 122 |
| County Controls | Yes | Yes | Yes | Yes | Yes | Yes | Yes | Yes | Yes |

Notes.—Sources: Department of Veterans Affairs, American Community Survey 2014–2018. The table reports the coefficients associated with regressions of an individual VA center's overall quality on a vector of characteristics, conditional on county demographic controls, including: logged population, the share married, the share male, the age distribution (the share under the age of 18, 18–24, 25–34, 35–44, 45–54, 65–74, 75–84, and 85+), the education distribution (the share with less than a high school degree, some college, and college or more). Our VA medical center characteristics are defined as follows: Overall Rating of Hospital: Patients who gave their hospital a rating of 9 or 10 on a scale from 0 (lowest) to 10 (highest), Communication with Nurses and Doctors: Patients who reported that their nurses "Always" communicated well and Nurses "always" communicated well, Care Transition: Patients who "Strongly Agree" they understood their care when they left the hospital and Patients who "Strongly Agree" they understood their care when they left the hospital, Cleanliness of the Hospital Environment: Patients who reported that their room and bathroom were "Always" clean and Room was "always" clean, Communication about Medication: Patients who reported that staff "Always" explained about medicines before giving it to them and Staff "always" explained, Pain Management: Patients who reported that their pain was "Always" well controlled and Pain was "always" well controlled, Quietness of the Hospital Environment: Patients who reported that the area around their room was "Always" quiet at night, Responsiveness of Hospital Staff: Patients who reported that they "Always" received help as soon as they wanted and Patients "always" received help as soon as they wanted, Willingness to Recommend Hospital: Patients who reported YES they would definitely recommend the hospital. Standard errors are clustered at the VA-hospital level and observations are unweighted.

channels through which VA medical centers affect Veteran outcomes under the PPP. In particular, we draw on additional data from 2016 on a sample of 122 VA medical centers, decomposing perceptions of overall quality.

Table 5 documents these results. We begin by regressing logged overall ratings on each characteristic independently, adding them all together in column 9, controlling for county demographic characteristics in each specification. While communication with doctors and nurses is most predictive when each of the characteristics are viewed in isolation (column 1), we find that willingness to recommend the VA medical center and hospital cleanliness are most important when we control for them all jointly, followed by whether the hospital is quiet. Interestingly, communication about medication is negatively correlated with overall quality, which could reflect the fat that centers with more medication are subject to more challenging medical cases.

What do these results imply about the mechanism through which VA medical centers might affect Veteran outcomes in the PPP? Although the willingness to recommend a VA medical center is admittedly a coarse proxy for quality, it fundamentally captures whether a

respondent had a positive experience and views the location as helpful for their flourishing as an individual. Importantly, the experience includes more than just medical treatment: it is also about relationship building and the acquisition of information. We now consider whether VA medical centers potentially serve as a hub for the dissemination of information and community building that is integral for successful entrepreneurship. This would represent a new way that VA medical centers can enhance the well-being of Veterans beyond bringing physical healing and treatment.

## Conclusion

There is increasing evidence that some of the funds from the PPP were allocated to firms that needed them less than others. One important explanation resides with the fact that the distribution of funds was heavily determined by the quality of banking relationships at a local-level prior to the pandemic [6]. Since the funds were distributed through the banking system, small business owners who did not have strong ties to local banks were unable to apply rapidly and properly. Another and closely related explanation is that information frictions made it difficult for small business owners to understand the offer and how to apply for PPP loans [8]. When the PPP was first launched, less than 70% were even aware of it. Moreover, small business owners with less than 10 employees were systematically less likely to be aware of the program. To the extent that higher quality VA medical centers operate in part as community centers where Veterans are able to cultivate relationships and exchange information, then it is possible that Veterans gained access to special information at the better medical centers.

In sum, our results show that Veterans fared better than their counterparts from the PPP. For example, we show that Veterans received 3.5% more loans and 6.8% larger loans than their counterparts. Our statistical strategy controls for a wide array of zipcode characteristics, such as the age and education distribution, and exploits within-zipcode variation in further robustness. That allows us to compare differences in PPP receipt among Veterans and non-Veterans within the same zipcode over time. We also show that the quality of local VA medical centers play an important role in accounting for these differences in PPP lending to Veterans, in addition to standard demographics about the zipcode that the medical center is located within.

Future research is needed to better understand the channels through which VA medical centers can advance local outcomes for Veterans. For example, VA medical centers could serve as a source for the dissemination of information. That is, they can be places where people come together to not only receive direct medical attention, but also best practices for entrepreneurship and financial literacy. Given that a big barrier towards receipt of the PPP was information frictions, better-run VA medical centers could lead to the diffusion of information about local lending opportunities.

## Supporting information

**S1 Appendix.**
(PDF)

## Acknowledgments

We thank Adi Bejan for detailed comments.

## Author Contributions

**Conceptualization:** Christos A. Makridis.

**Data curation:** Christos A. Makridis, J. D. Kelly.

**Formal analysis:** Christos A. Makridis, J. D. Kelly.

**Investigation:** Christos A. Makridis.

**Methodology:** Christos A. Makridis.

**Project administration:** Christos A. Makridis.

**Software:** Christos A. Makridis.

**Supervision:** Christos A. Makridis, Gil Alterovitz.

**Validation:** Christos A. Makridis, J. D. Kelly.

**Visualization:** Christos A. Makridis, J. D. Kelly.

**Writing – original draft:** Christos A. Makridis, J. D. Kelly.

**Writing – review & editing:** Christos A. Makridis, Gil Alterovitz.

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
