## [Decision Letter · Decision Letter 0]

14 Mar 2022

PONE-D-21-24369The Effects of Medical Centers on Socio-economic Outcomes: Evidence from the Paycheck Protection ProgramPLOS ONE

Dear Dr. Makridis,

Thank you for submitting your manuscript to PLOS ONE. After careful consideration, we feel that it has merit but does not fully meet PLOS ONE’s publication criteria as it currently stands. Therefore, we invite you to submit a revised version of the manuscript that addresses the points raised during the review process.

Please consider all comments.

We look forward to receiving your revised manuscript.

Kind regards,

Ahmed Mancy Mosa, Ph.D.

Academic Editor

PLOS ONE

Journal Requirements:

4. Thank you for stating the following financial disclosure: "The funders had no role in study design, data collection and analysis, decision to publish, or preparation of the manuscript."

Reviewers' comments:

Reviewer's Responses to Questions

**Comments to the Author**

1. Is the manuscript technically sound, and do the data support the conclusions?

Reviewer #1: Yes

Reviewer #2: Partly

2. Has the statistical analysis been performed appropriately and rigorously? 

Reviewer #1: Yes

Reviewer #2: Yes

3. Have the authors made all data underlying the findings in their manuscript fully available?

Reviewer #1: Yes

Reviewer #2: Yes

4. Is the manuscript presented in an intelligible fashion and written in standard English?

Reviewer #1: Yes

Reviewer #2: Yes

5. Review Comments to the Author

Reviewer #1: The reviewed article raises an interesting and timely issue - the impact of the global Covid-19 pandemic on veterans, a group particularly vulnerable to the effects of the pandemic due to their age. The work is quite carefully done, although I have a few comments:

1) the title is too general and does not reflect the scope of the work. It should be made clear that the study is about veterans, as the current form promises a broader scope of content

2) phrases like "While we are not the first to study the Paycheck Protection Program" sound too colloquial for an academic paper

Other than that, I appreciate the originality of the research problem chosen by the authors of the paper.

Reviewer #2: The paper investiagtes the impact of access to medical care on economic outcomes.

The analysis regards the group of Veterans and is carried out during COVID-19 pandemic.

The results show that Veterans received more loans and larger loans compared to their counterparts.

The study is interesting and well-motivated however it needs more explanations at some points.

Major issues:

1. The literature review lacks studies investigating the relations between healthcare

services and economic performance. While this relation is rather not straightforward and not intuitive

I encourage authors to enrich this section.

2. You write that about 80% of race information and 75% of status is missing in the dataset.

So maybe the dataset is not representative. E.g. maybe PPP loans are more likely given to

the wealthier and white veterans as they might be overrepresented in your dataset?

3. You don't sufficiently explain why PPP loans are more likely given to veterans?

Please provide a logical explanation taking into account your results.

In my opinion, you overestimate the role of VA Medical Centers in the development of social capital.

E.g. the visits at these facilities are not so regular to create social ties. Moreover, the primary aim is healthcare, not chats.

Am I wrong?

4. What is the relation between the VA Medical Centers ratings and PPP loans?

As I don't see the connection, please provide a logical explanation.

6. PLOS authors have the option to publish the peer review history of their article (what does this mean?). If published, this will include your full peer review and any attached files.

Reviewer #1: No

Reviewer #2: No

---

## [Decision Letter · Decision Letter 1]

25 May 2022

The Effects of Department of Veterans Affairs Medical Centers on Socio-economic Outcomes: Evidence from the Paycheck Protection Program

PONE-D-21-24369R1

Dear Dr. Makridis,

We’re pleased to inform you that your manuscript has been judged scientifically suitable for publication and will be formally accepted for publication once it meets all outstanding technical requirements.

Kind regards,

Ahmed Mancy Mosa, Ph.D.

Academic Editor

PLOS ONE

Additional Editor Comments (optional):

Reviewers' comments:

Reviewer's Responses to Questions

**Comments to the Author**

1. If the authors have adequately addressed your comments raised in a previous round of review and you feel that this manuscript is now acceptable for publication, you may indicate that here to bypass the “Comments to the Author” section, enter your conflict of interest statement in the “Confidential to Editor” section, and submit your "Accept" recommendation.

Reviewer #2: All comments have been addressed

2. Is the manuscript technically sound, and do the data support the conclusions?

Reviewer #2: Yes

3. Has the statistical analysis been performed appropriately and rigorously? 

Reviewer #2: Yes

4. Have the authors made all data underlying the findings in their manuscript fully available?

Reviewer #2: No

5. Is the manuscript presented in an intelligible fashion and written in standard English?

Reviewer #2: Yes

6. Review Comments to the Author

Reviewer #2: The changes made by authors in this version of the paper are rather symbolic. However, the study presents coherent analysis of the outlined problem. This version is ok.

7. PLOS authors have the option to publish the peer review history of their article (what does this mean?). If published, this will include your full peer review and any attached files.

Reviewer #2: No

---

## [Editor Report · Acceptance letter]

27 Jul 2022

PONE-D-21-24369R1 

The Effects of Department of Veterans Affairs Medical Centers on Socio-economic Outcomes: Evidence from the Paycheck Protection Program 

Dear Dr. Makridis:

I'm pleased to inform you that your manuscript has been deemed suitable for publication in PLOS ONE. Congratulations! Your manuscript is now with our production department. 

Kind regards, 

on behalf of

Dr. Ahmed Mancy Mosa 

Academic Editor

PLOS ONE